## [Peer Review File · EMBO Molecular Medicine]

A bactericidal tuberculosis drug regimen driven by inhibition of the terminal oxidases by pretomanid

Nurlilah Rahman, Samsher Singh, Thomas Wiggins, May Santos, Garrett Moraski, Marvin Miller, Michael Berney, and Kevin Pethe

Corresponding author: Kevin Pethe (kevin.pethe@biology.ox.ac.uk)

Review Timeline:

Submission Date:	9th Jan 25
Editorial Decision:	28th Jan 25
Revision Received:	1st Oct 25
Editorial Decision:	18th Nov 25
Revision Received:	17th Jan 26
Accepted:	20th Jan 26

Editor: Zeljko Durdevic

Transaction Report:

28th Jan 2025

Dear Dr. Pethe,

Thank you for the submission of your manuscript to EMBO Molecular Medicine. We have now received feedback from the three reviewers who agreed to evaluate your manuscript. All three referees recognize interest of the study but also raise important concerns that should be addressed in a major revision. Focus of the revision should be in assessing the discrepancies between in vitro and in vivo experiments and the in vivo resistance development. Given that the revision will require extensive experimentation we think six months rather than three months would be more appropriate to provide the complete revision. If you would like to discuss further the points raised by the referees, I am available to do so via email or video. Let me know if you are interested in this option.

We would welcome the submission of a revised version within six months for further consideration. Please let us know if you require longer to complete the revision.

I look forward to receiving your revised manuscript.

Yours sincerely,

Zeljko Durdevic

We require:

- 1) A .docx formatted version of the manuscript text (including legends for main figures, EV figures and tables). Please make sure that the changes are highlighted to be clearly visible.
- 2) Individual production quality figure files as .eps, .tif, .jpg (one file per figure). For guidance, download the 'Figure Guide PDF': (<https://www.embopress.org/page/journal/17574684/authorguide#figureformat>).
- 3) A .docx formatted letter INCLUDING the reviewers' reports and your detailed point-by-point responses to their comments. As part of the EMBO Press transparent editorial process, the point-by-point response is part of the Review Process File (RPF), which will be published alongside your paper.
- 4) A complete author checklist, which you can download from our author guidelines (<https://www.embopress.org/page/journal/17574684/authorguide#submissionofrevisions>). Please insert information in the checklist that is also reflected in the manuscript. The completed author checklist will also be part of the RPF.
- 5) Please note that all corresponding authors are required to supply an ORCID ID for their name upon submission of a revised manuscript.

6) It is mandatory to include a 'Data Availability' section after the Materials and Methods. Before submitting your revision, primary datasets produced in this study need to be deposited in an appropriate public database, and the accession numbers and database listed under 'Data Availability'. Please remember to provide a reviewer password if the datasets are not yet public (see <https://www.embopress.org/page/journal/17574684/authorguide#dataavailability>).

.

- the medical issue you are addressing,

- the results obtained and

- their clinical impact.

12) Author contributions: You will be asked to provide CRediT (Contributor Role Taxonomy) terms in the submission system. These replace a narrative author contribution section in the manuscript.

13) A Conflict of Interest statement should be provided in the main text.

14) Every published paper now includes a 'Synopsis' to further enhance discoverability. Synopses are displayed on the journal webpage and are freely accessible to all readers. They include a short stand first (maximum of 300 characters, including space)

as well as 2-5 one-sentences bullet points that summarizes the paper. Please write the bullet points to summarize the key NEW findings. They should be designed to be complementary to the abstract - i.e. not repeat the same text. We encourage inclusion of key acronyms and quantitative information (maximum of 30 words / bullet point). Please use the passive voice. Please attach these in a separate file or send them by email, we will incorporate them accordingly.

15) Include a Reagents and Tools Table as part of the Methods section, which can be downloaded from our author guidelines (<https://www.embopress.org/page/journal/17574684/authorguide#structuredmethods>)

**** Reviewer's comments ****

Referee #1 (Comments on Novelty/Model System for Author):

The mouse experiments need additional data. The bacterial loads at the beginning of treatment (2 weeks post infection) need to be shown. Without these data it is unclear if the drugs had static or cidal effects, the how much killing was achieved. It is unclear how many data points/ replicates were included in Figure 4 D. Please show the individual data points.

Referee #1 (Remarks for Author):

This is an interesting manuscript providing insights in the suitability of combining tuberculosis drugs that target respiration. The in vitro data clearly demonstrate the synergy between pretomanid (PMD) and telacebec (Q203) and the cytochrome bd oxidase inhibitor ND-011992. While the ATP and OCR measurements implicate an effect of pretomanid on respiration, it remains unclear if this is a direct or an indirect effect. Experiments addressing causality are missing. Notwithstanding the observed synergies are very interesting and the impact on viability of Mtb is substantial. The impact during mouse infection is less drastic and raises concerns. While PMD and Q203 synergize well in vitro leading to significantly more killing than either drug alone, the combined effect in mouse lungs is modest (less than a log). In fact, it is unclear if any substantial killing was achieved, because the bacterial burden at the onset of treatment, i.e. 2 weeks post infection, has not been reported. These data must be added. The impact of PMD against H37Rv Δ cydAB is also modest in vivo and addition of PMD to Q203 increased the killing from Q203 by ~10-fold. It would be helpful and supporting the idea that combining PMD with Q203 and a bd oxidase inhibitor is a worthwhile strategy for treating TB, if the authors would have assessed resistance development and relapse in vivo. In the current manuscript the in vitro and in vivo experiments are somewhat inconsistent. It is also informative to assess the frequency of resistance mutants for the individual drugs and drug combinations formally, using a fluctuation assay.

Additional specific comments:

In Fig. 3b the impact of Q203 on the activity of PMD is only evident after 15 days. Can the authors explain or speculate why this impact is so delayed.

Fig. 4 D, please clarify how many replicates were included and show individual data points.

Fig. 5. CFU at start of treatment must be included. And please also show spleen data to assess if treatment is sufficient to prevent/reduce dissemination.

Referee #2 (Comments on Novelty/Model System for Author):

Data were analyzed using appropriate statistical methods.

All experiments using M.tb were conducted within a certified BSL3 containment condition.

Referee #2 (Remarks for Author):

The manuscript entitled "A bactericidal tuberculosis drug regimen driven by inhibition of the terminal oxidases by pretomanid" seeks to uncover the molecular targets of pretomanid (PMD) by monitoring the antimicrobial effects on M.tb or cydAB KO in a replicating or PBS starved non-replicating state after combination treatment with Q203 and/or ND-011992. Molecular target in M.tb bioenergetic activity of PMD was assumed because Q203 and ND-011992 target cyt bcc:aa3 oxidase and cyt bd oxidase of M. tb ETC, respectively. This study provides the most effective combination of other anti-TB agents targeting M. tb ETC terminal oxidases to improve the antimicrobial effect and curtail the emergence of drug resistance against PMD in vitro and in murine model.

Overall, the study is invaluable in that the results represent new antibiotic regimen including PMD to kill M.tb in a replicating and a non-replicating state. Despite of the strength, the reviewer has some concerns regarding the data analysis and interpretation. No direct experimental evidence was shown that PMD targets M.tb cytochrome bcc:aa3 oxidase and bd oxidase in Mtb electron transport chain. In a previous study and the result in this study showed that antibiotics targeting Mtb mycolic acid synthesis showed intrabacterial ATP surge, which is antagonistic in antimicrobial effects against M.tb. In Fig. 1A, PMD in a range between 0.1 to 5 uM increased Mtb ATP content, suggesting that PMD under these concentrations target Mtb mycolic acid biosynthesis. Authors use 1.2 - 5 uM PMD in all experiments against PBS starved non-replicating Mtb and replicating Mtb, which is thought to kill M.tb by targeting Mtb cell wall biogenesis.

Throughout the study, authors forgot describing if the study uses M.tb bacilli replicating or non-replicating state. Also, many figures don't match the symbols of the graph lines to the figure legends.

Fig. 1A: ATP content intracellular Mtb after treatment with various concentrations of three antibiotics such as PMD, INH, and BDQ. The level of intracellular ATP content should be displayed together with the M.tb killing curves under the same concentrations of each Abx. This will provide the functional link between the intracellular bioenergetic response and different phenotypic state.

Page 3, L14: It is required to provide the definition what does it mean by low and high concentrations of PMD by providing the M.tb phenotypic response in parallel. Explain whether the abrogated ATP surge in high PMD concentrations is due to M.tb metabolic shift towards the non-replicating state or just arising from the dead effects.

Fig. 1B: It is unclear whether the % change of ATP level is relative to the replicating counterpart or that before treatment with the antibiotic. It should make it clear because intrabacterial ATP of Mtb in a non-replicating state maintains at levels significantly lower than those in a replicating state.

Fig. 2A: it is unclear if authors used PBS starved non-replicating M.tb for the measurement of impact of Abx on OCR. Page 4, Ln 1: This is referred to the Lamprecht et al., 2016, where OCR was measured in M.tb in a replicating state.

Fig. 2B: Is it the result of cydAB KO OCR? As cydAB KO in a PBS starved condition maintains OCR at a similar level to WT (Fig. 2A), OCR reduction of cydAB KO in response to treatment with Q203 or PMD indicated that the targets of PMD share with that of Q203, which is cytochrome bcc:aa3 not bd oxidase?

Fig. 3: In figure legends, authors use 2.5 and 5 uM PMD, where intracellular ATP content is surged, the evidence showing that PMD at those concentrations targets M.tb cell wall biosynthesis, not M.tb membrane bioenergetics.

Page 4, Ln17: check the following sentence - demonstrating that inhibition of the cyt-bcc:aa3 potentiates the bactericidal potency of pretomanid against M.tb in a non-replicating state.

Fig. 3E: Symbols do not match the condition in the graph.

Page 4. Ln 28: Authors seek to explain the antimicrobial synergy of intensive inhibition of ETC terminal oxidases with pretomanid, when it targets membrane bioenergetics. The concentrations of pretomanid used in Fig. 3 were at around 1.5 uM, in which intracellular ATP content surged and normally shows antagonistic effects with anti-TB agents targeting M.tb membrane bioenergetics.

Fig. 4A, Page 5. Ln 3: by day 10 post-treatment. PMD treatment initially killed M.tb in a replicating state during the first 3 days and the bacilli resumed its growth, reaching the density similar to initial input by day 10 PT (or untreated control by day 20 PT).

Fig. 5A: Based upon the Q203 CFU compared to that of vehicle, M.tb in mice may still be in a replicating state, similar to the result in Fig. 4A. However, the antimicrobial synergy of PMD and Q203 in vivo was significantly weaker than that shown in vitro. Authors may explain the possible reason in the discussion section.

Minor

In text, authors keep using Abx name, telecebec, but in figures, authors use Q203. It's better to use same name.

Referee #3 (Remarks for Author):

The authors, Rahman et al., describe an important finding on Pretomanid. Pretonamid is a new anti-tuberculosis medication that works in two ways: it weakens the bacteria's outer layer and disrupts its energy production. They show that pretomanid targets two specific parts of the bacteria's energy system, making it more effective and less likely to lead to drug resistance. Combining pretomanid with other drugs that target the same energy pathways could create a powerful treatment for tuberculosis, especially against hard-to-treat forms of the disease. The experiments are well-planned, executed, and analyzed. The study significantly contributes to the tuberculosis drug-discovery efforts. However, I have a few suggestions to improve this MS.

- It is unclear why authors are only considering MIC50 and not MIC90 - a clarification is needed, especially considering the translation of MIC50 in in vivo efficacy studies.

- For the emergence of drug-resistance experiment, Fig 4B, I am wondering whether the authors have validated the resistance genotypically; based on the data the observed growth of bacterial cells on the pretomanid-containing plate could be due to a variety of reasons.

- For the in vivo experiment, it is unclear which mice model was used; is it a chronic model to compare the effect of (observed in vitro cidality on non-replicating nutrient-starved culture) combination of drugs? Also, a detailed PK would have been useful to compare the in vitro and in vivo data.

Referee #1 (Comments on Novelty/Model System for Author)

- The mouse experiments need additional data. The bacterial loads at the beginning of treatment (2 weeks post-infection) need to be shown. Without these data, it is unclear if the drugs had static or cidal effects, and how much killing was achieved.

We agree with this point and included bacterial loads at two weeks post-infection in Figure 5A and 5B.

- It is unclear how many data points/replicates were included in Figure 4D. Please show the individual data points.

Figure 4D legend: “replicate” was changed to “triplicate”. The format of the figure was changed to show the individual data points.

Referee #1 (Remarks for Author)

- While the ATP and OCR measurements implicate an effect of pretomanid on respiration, it remains unclear if this is a direct or an indirect effect. Experiments addressing causality are missing.

We acknowledge this concern. The impact on pretomanid the electron transport chain is mediated through nitric oxide (NO) release following bioactivation by *ddn*, with F420 acting as cofactor. This mechanism complicates direct biochemical assays, such as those with inverted membrane vesicles (IMVs), because of the inherent instability of NO. In addition, the requirement for multiple components, purified Ddn, reduced F420 (which itself can influence IMV respiration), renders such assays technically challenging and difficult to control.

To address this, we adopted an alternative strategy. We employed a nitric oxide donor as a proxy for pretomanid and tested its effect on purified mycobacterial IMVs. Since *M. tuberculosis* IMVs cannot be purified due to biosafety constraints, we used IMVs from *M. smegmatis* as a surrogate system. Using this approach, we demonstrated that NO inhibits both the cytochrome *bcc:aa₃* and cytochrome *bd* respiratory branches. These results have now been incorporated as Supplementary Figure 1, and the corresponding description has been added to the Results section.

- The impact during mouse infection is less drastic and raises concerns. While PMD and Q203 synergize well *in vitro*, leading to significantly more killing than either drug alone, the combined effect in mouse lungs is modest (less than a log). In fact, it is unclear if any substantial killing was achieved, because the bacterial burden at the onset of treatment, i.e., 2 weeks post-infection, has not been reported. These data must be added.

We have now included CFU counts at the start of treatment to provide a clearer basis for interpreting the impact of the drug combination *in vivo*. The reduction in bacterial load observed with the Q203+pretomanid combination is statistically significant against the parental strain. We acknowledge that the *in vivo* effects appear less pronounced than the *in vitro* results, which could be due to several factors, including: (i) the use of the H37Rv strain, which expresses higher levels of *cyt-bd* compared with more recent clinical isolates (DOI: 10.1128/AAC.03486-14); and (ii) challenges in defining appropriate doses for two-drug combinations.

Importantly, the strong efficacy of the Q203+pretomanid combination against the H37Rv Δ cydAB mutant supports the concept that triple regimens comprising pretomanid, Q203, and a *cyt-bd* inhibitor represent a promising therapeutic strategy. We have highlighted this more explicitly in the Discussion, emphasizing the need to develop potent *cyt-bd* inhibitors with *in vivo* efficacy. In addition, we have included new data showing the potency of the Q203-pretomanid combination against the H37Rv Δ cydAB strain in both lung and spleen after 6 weeks of treatment.

- It would be helpful and supporting the idea that combining PMD with Q203 and a *bd* oxidase inhibitor is a worthwhile strategy for treating TB if the authors would have assessed resistance development and relapse *in vivo*.

We agree that an *in vivo* relapse experiment would provide valuable insights. However, our laboratory has limited resources to conduct such studies, which require trained personnel, substantial amounts of drug for long-term regimens, and appropriate facilities. As an alternative, we designed a new experiment using the H37Rv Δ cydAB strain to evaluate the time needed to reduce lung and spleen CFUs below the limit of detection (LOD). By increasing the pretomanid dose to 75 mg/kg while keeping Q203 at 10 mg/kg, we found that 6 weeks of treatment was sufficient to reduce bacterial burdens in both lung and spleen to LOD levels. While Q203 alone was able to sterilize the lungs, addition of pretomanid was required to achieve the same outcome in the spleen. Notably, a positive interaction between pretomanid and Q203 was already evident in the lung after 3 weeks of treatment. These data have been added to Figure 5 and as supplementary Figure 2.

We agree that relapse experiments will be particularly informative once highly potent *cyt-bd* inhibitors with favourable pharmacokinetic properties become available. To address the question of resistance, we plated organ homogenates on pretomanid-containing agar. No resistant colonies were recovered, which may be a consequence of the low bacterial counts in mice and/or the possibility that common *in vitro* loss-of-function mutations carry a fitness cost *in vivo*. This point is discussed in the revised manuscript.

It is also informative to assess the frequency of resistance mutants for the individual drugs and drug combinations formally, using a fluctuation assay.

A fluctuation analysis assay was performed; it confirmed that the combination of Q203-pretomanid limited emergence of resistance to both Q203 and pretomanid. Results were added as Supplementary table 2.

- In Fig. 3B, the impact of Q203 on the activity of PMD is only evident after 15 days. Can the authors explain or speculate why this impact is so delayed?

We hypothesize that the delayed effect may reflect the time required for cellular stress to accumulate following inhibition of multiple respiratory components, although the precise mechanism remains unclear. It is noteworthy that other respiratory inhibitors, such as bedaquiline and telacebec, also display slow onset of action, with limited bactericidal activity observed during the first 7–10 days of *in vitro* incubation. The underlying reasons for this delayed killing remain to be elucidated.

- Fig. 5: CFU at the start of treatment must be included. Also, please show spleen data to assess if treatment is sufficient to prevent/reduce dissemination.

CFU data at the start of treatment were added. We did not perform spleen counts in the initial experiments but included the counts in the new experiment using the H37Rv Δ cydAB strain (Figure 5 and suppl. Figure 2)

Referee #2 (Comments on Novelty/Model System for Author)

- No direct experimental evidence was shown that PMD targets M.tb cytochrome bcc:aa3 oxidase and bd oxidase in M.tb electron transport chain.

We acknowledge this limitation. Please see reply to reviewer 1 on a similar comment.

Throughout the study, authors forgot to describe whether the study uses M.tb in a replicating or non-replicating state.

Each figure and results now explicitly states whether the data correspond to replicating or non-replicating Mtb.

- Fig. 1A: The level of intracellular ATP content should be displayed together with the M.tb killing curves under the same concentrations of each antibiotic. This is a useful suggestion.

The issue we are facing is the delayed killing relative to the effect on ATP levels. ATP depletion is rapid, usually visible in 6 to 12 hours, but the effect on bacterial death is much delayed, only becoming apparent after 7-10 days, sometimes faster using drug combination but usually taking days rather than hours.

- Page 3, L14: It is required to provide the definition of what is meant by low and high concentrations of PMD.

We changed this imprecise nomenclature and added the concentration of PMD used

- Fig. 5A: The antimicrobial synergy of PMD and Q203 in vivo was significantly weaker than that shown in vitro. Authors may explain the possible reason in the discussion section.

We expanded the discussion on this discrepancy, considering factors such as drug pharmacokinetics, drug dose, etc in response to this comment as well as the comment from reviewer 1

- Minor: Authors use the name "telecebec" in the text but "Q203" in the figures. It's better to use a consistent name.

We standardized the nomenclature throughout the manuscript to avoid confusion.

Referee #3 (Remarks for Author)

- It is unclear why authors are only considering MIC₅₀ and not MIC₉₀ - a clarification is needed, especially considering the translation of MIC₅₀ in in vivo efficacy studies.

We determine MIC₅₀ based on a full dose-response curve because it usually gives a more robust and reproducible value, very much like enzymologists use IC₅₀ values. MIC₉₀ is more variable due to its dependence on the Hill slope of the dose-response curve, which can lead to reproducibility issues, especially for

drugs such as telacebec that possess a shallow hill-slop. We used MIC₅₀ in all of our publications.

- For the emergence of drug-resistance experiment (Fig. 4B), I am wondering whether the authors have validated the resistance genotypically; based on the data, the observed growth of bacterial cells on the pretomanid-containing plate could be due to a variety of reasons.

We have isolated some escape mutants, confirmed their level of resistance to PMD, and showed the emergence of single-nucleotide polymorphisms in genes previously associated with resistance to pretomanid. Results were added as Supplementary Table 1.

- For the *in vivo* experiment, it is unclear which mouse model was used. Is it a chronic model to compare the effect of (observed *in vitro* cidality on non-replicating nutrient-starved culture) combination of drugs? Also, a detailed PK would have been useful to compare the *in vitro* and *in vivo* data.

We used an acute infection model for the first *in vivo* studies (Figure 5 A and B), starting treatment at 2 weeks post-infection. For the second experiment, we used a more established model, starting treatment at 4 weeks post-infection. We made it clearer in the revised manuscript, especially in the Figure 5 legend. Pharmacokinetics of both pretomanid and Q203 are already known, we did not feel the need to repeat those experiments. Since both drugs are not CYP450 activators, their PK profile should be independent. Results using the H37rv Δ cydAB (initial data as well as new data figure 5C and 5D) indicate that both PMD and Q203 have high potency alone, indicating that they reach drug levels in blood and spleen at level high enough to achieve efficacy reported by other research groups.

18th Nov 2025

Dear Dr. Pethe,

Thank you for the submission of your manuscript to EMBO Molecular Medicine. I am pleased to inform you that we will be able to accept your manuscript pending the following final amendments:

1) Please implement all referees' suggestions.

2) Figures:

- Remove all figures from the main manuscript file.

- Upload supplementary figures as EV figures and rename them to "Figure EV1" etc. Please update their callouts in the main text.

- Please upload main figures and EV figures as individual, high-resolution files in TIFF, EPS or PDF format. The legends should be compiled at the end of the manuscript text, with the EV figure legends after the main figure legends and with the heading "Expanded View Figure Legends" Please check "Author Guidelines" for more information:

<https://www.embopress.org/page/journal/17574684/authorguide#figureformat>

<https://www.embopress.org/page/journal/17574684/authorguide#expandedview>

3) In the main manuscript file, please do the following:

- Please address all comments suggested by our data editors listed below:

o Figure legends:

1. Please note that the exact p values are not provided in the legends of figures 2A-C; 3A-C, E; 4D, 5A-D; S2 A.

2. Please note that information related to n is missing in the legends of figures 1A, 2A-C; 3A-E.

3. Please note that the error bars are not defined in the legends of figures 1A, B; 2A-C; 3A-E; 4A-C.

- Add up to 5 keywords.

- Please include structured Methods section that includes a Reagents and Tools Table followed by a Methods and Protocols section. More information on how to adhere to this format as well as downloadable templates (.docx) for the Reagents and Tools Table can be found in our author guidelines: <https://www.embopress.org/page/journal/17574684/authorguide#structuredmethods>
An example of a paper with Structured Methods can be found here:

<https://www.embopress.org/doi/full/10.1038/s44320-024-00037-6#sec-4>

- Rename "Material and Methods" to "Methods".

- In Methods, add statistical paragraph that should reflect all information that you have filled in the Authors Checklist, especially regarding randomization, blinding, replication etc.

- Indicate in legends exact n and exact p values, not a range, along with the statistical test used. To keep the figures "clear" some authors found providing an Appendix table Sx with all exact p-values preferable. You are welcome to do this if you want to.

- Remove BioRender information from the Acknowledgements and add a dedicated section to the Methods following this format:
Graphics:

(some of the... OR Figure #... OR synopsis) Graphics were created with BioRender.com.

- Add "Disclosure Statement & Competing Interests". We updated our journal's competing interests policy in January 2022 and request authors to consider both actual and perceived competing interests. Please review the policy <https://www.embopress.org/competing-interests> and update your competing interests if necessary.

- Author contributions: Please remove it from the manuscript and specify author contributions in our submission system. CRediT has replaced the traditional author contributions section because it offers a systematic machine-readable author contributions format that allows for more effective research assessment. You are encouraged to use the free text boxes beneath each contributing author's name to add specific details on the author's contribution. More information is available in our guide to authors:

<https://www.embopress.org/page/journal/17574684/authorguide#authorshipguidelines>

4) Tables: Please upload supplementary tables as separate EV table excel files, rename them to Table EV1 etc. with their legends in the same sheet and their callouts updated in the main text.

5) Synopsis: Please check your synopsis text and image before submission with your revised manuscript. Please be aware that in the proof stage minor corrections only are allowed (e.g., typos).

6) As part of the EMBO Publications transparent editorial process initiative (see our Editorial at <http://embomolmed.embopress.org/content/2/9/329>), EMBO Molecular Medicine will publish online a Review Process File (RPF) to accompany accepted manuscripts. This file will be published in conjunction with your paper and will include the anonymous referee reports, your point-by-point response and all pertinent correspondence relating to the manuscript. Let us know whether you agree with the publication of the RPF and as here, if you want to remove or not any figures from it prior to publication. Please note that the Authors checklist will be published at the end of the RPF.

7) Please provide a point-by-point letter INCLUDING my comments as well as the reviewer's reports and your detailed responses (as Word file).

I look forward to reading a new revised version of your manuscript as soon as possible.

Yours sincerely,

Zeljko Durdevic

Zeljko Durdevic
Senior Editor
EMBO Molecular Medicine

*** Instructions to submit your revised manuscript ***

- 1) a .docx formatted version of the manuscript text (including Figure legends and tables)
- 2) Separate figure files*
- 3) supplemental information as Expanded View and/or Appendix. Please carefully check the authors guidelines for formatting Expanded view and Appendix figures and tables at <https://www.embopress.org/page/journal/17574684/authorguide#expandedview>
- 4) a letter INCLUDING the reviewer's reports and your detailed responses to their comments (as Word file).
- 5) The paper explained: EMBO Molecular Medicine articles are accompanied by a summary of the articles to emphasize the major findings in the paper and their medical implications for the non-specialist reader. Please provide a draft summary of your article highlighting
 - the medical issue you are addressing,
 - the results obtained and
 - their clinical impact.This may be edited to ensure that readers understand the significance and context of the research. Please refer to any of our published articles for an example.
- 6) Author contributions: the contribution of every author must be specified in our submission system.
- 7) EMBO Molecular Medicine now requires a complete author checklist (<https://www.embopress.org/page/journal/17574684/authorguide>) to be submitted with all revised manuscripts. Please use the checklist as guideline for the sort of information we need WITHIN the manuscript. The checklist should only be filled with page numbers where the information can be found. This is particularly important for animal reporting, antibody dilutions (missing) and exact values and n that should be indicated instead of a range.
- 8) Every published paper now includes a 'Synopsis' to further enhance discoverability. Synopses are displayed on the journal webpage and are freely accessible to all readers. They include a short stand first (maximum of 300 characters, including space) as well as 2-5 one sentence bullet points that summarise the paper. Please write the bullet points to summarise the key NEW findings. They should be designed to be complementary to the abstract - i.e. not repeat the same text. We encourage inclusion of key acronyms and quantitative information (maximum of 30 words / bullet point). Please use the passive voice. Please attach these in a separate file or send them by email, we will incorporate them accordingly.

You are also welcome to suggest a striking image or visual abstract to illustrate your article. If you do please provide a jpeg file

550 px-wide x 300-600px high.

9) A Disclosure Statement & Competing Interests statement should be provided in the main text

10) Please note that we now mandate that all corresponding authors list an ORCID digital identifier. This takes <90 seconds to complete. We encourage all authors to supply an ORCID identifier, which will be linked to their name for unambiguous name identification.

Currently, our records indicate that the ORCID for your account is 0000-0003-0916-8873.

Link Not Available

11) Include a Reagents and Tools Table as part of the Methods section, which can be downloaded from our author guidelines (<https://www.embopress.org/page/journal/17574684/authorguide#structuredmethods>)

Photos 400-800 DPI

*Additional important information regarding figures and illustrations can be found at

<https://bit.ly/EMBOPressFigurePreparationGuideline>. See also figure legend preparation guidelines:

<https://www.embopress.org/page/journal/17574684/authorguide#figureformat>

***** Reviewer's comments *****

Referee #1 (Comments on Novelty/Model System for Author):

My rationale for scoring "medium" for novelty is that it is well known that Q203 is highly active against mycobacteria lacking bd oxidase (e.g. *M. leprae*, *M. ulcerans*) and that PMD releases nitric oxide, presumably inhibiting respiration. While the authors demonstrate that NO can inhibit respiration, there is still no causal link that PMD inhibits respiration directly via NO generation. I think this limitation should at least be discussed.

Referee #1 (Remarks for Author):

Thank you for carefully addressing my concerns. There is still a lack of direct evidence that PMD inhibits respiration via NO generation. I understand the technical challenges, but this limitation should be discussed.

Referee #3 (Remarks for Author):

Authors, Rahman et al., have incorporated most of the suggestions, and the MS looks good. I agree with the limitations in verifying/performing certain experiments/data. However, I suggest that authors clarify whether they have used activated charcoal in the solid growth media for CFU enumeration - to avoid any compound carryover effect, which is very common in combination assays (and yields false positive results).

In Figure 5, while the legend provides the information, it is advised to include the experimental scheme (or highlight acute vs chronic in the Figure) and label drugs' concentrations for clarity. It would be good to have the Chronic data for H37Rv in the same Figure to compare with panels C and B (as in panel A, comparing with panel B).

***** Reviewer's comments *****

Referee #1 (Comments on Novelty/Model System for Author):

My rationale for scoring "medium" for novelty is that it is well known that Q203 is highly active against mycobacteria lacking bd oxidase (e.g. *M. leprae*, *M. ulcerans*) and that PMD releases nitric oxide, presumably inhibiting respiration. While the authors demonstrate that NO can inhibit respiration, there is still no causal link that PMD inhibits respiration directly via NO generation. I think this limitation should at least be discussed.

Referee #1 (Remarks for Author):

Thank you for carefully addressing my concerns. There is still a lack of direct evidence that PMD inhibits respiration via NO generation. I understand the technical challenges, but this limitation should be discussed.

We agree that the direct link is somewhat missing due to the unique mode of action of the drug. We added the following sentence in the discussion: "Although we did not directly quantify pretomanid-derived nitric oxide under our experimental conditions, the pattern of partial inhibition across both respiratory branches remains most consistent with an NO-mediated mechanism acting at the level of the heme-containing terminal oxidases."

Referee #3 (Remarks for Author):

Authors, Rahman et al., have incorporated most of the suggestions, and the MS looks good. I agree with the limitations in verifying/performing certain experiments/data. However, I suggest that authors clarify whether they have used activated charcoal in the solid growth media for CFU enumeration - to avoid any compound carryover effect, which is very common in combination assays (and yields false positive results).

Response: We thank the reviewer for raising this important point. Activated charcoal was not used in the solid media in this study. In prior unpublished observations, we found that, in contrast to bedaquiline, activated charcoal does not bind telacebec (Q203) on agar plates and therefore does not mitigate potential carryover effects for this compound.

Instead, we directly assessed drug carryover by plating *M. tuberculosis* immediately after exposure to the highest concentrations of pretomanid and telacebec used in the kill-kinetic experiments, corresponding to the maximal amount of drug that could be transferred onto agar plates in a 50 μ L inoculum. Under these conditions, no reduction in CFU was observed compared with untreated control cultures, indicating that any residual drug carried over was rapidly diluted in the agar and did not inhibit bacterial outgrowth.

These results demonstrate that neither pretomanid nor telacebec, alone or in combination, exhibited a measurable carryover effect under our experimental conditions. This control is now included as Appendix Fig. S1, and the following statement has been added to the main text: "Q203 or PMD alone, as well as their combination (Q203 + PMD), did not exhibit a carryover effect (Appendix Fig. S1)."

In Figure 5, while the legend provides the information, it is advised to include the experimental scheme (or highlight acute vs chronic in the Figure) and label drugs' concentrations for clarity. It would be good to have the Chronic data for H37Rv in the same Figure to compare with panels C and B (as in panel A, comparing with panel B).

Response: As suggested by the reviewer, we have now specified the concentrations of PMD and Q203 used in all experiments. The concentrations used for each drug alone, as well as the same concentrations used in combination, are indicated directly in the corresponding figure panels and described in the figure legends. In addition, the figure legends have been revised to clearly distinguish between the acute and established tuberculosis models.

20th Jan 2026

Dear Dr. Pethe,

We are pleased to inform you that your manuscript is accepted for publication and is now being sent to our publisher to be included in the next available issue of EMBO Molecular Medicine.

You may qualify for financial assistance for your publication charges - either via a Springer Nature fully open access agreement or an EMBO initiative. Check your eligibility: <https://link.springer.com/journal/44321/how-to-publish-with-us>

Zeljko Durdevic
Senior Editor
EMBO Molecular Medicine

>>> Please note that it is EMBO Molecular Medicine policy for the transcript of the editorial process (containing referee reports and your response letter) to be published as an online supplement to each paper. If you do NOT want this, you will need to inform the Editorial Office via email immediately. More information is available here: <https://link.springer.com/partners/embo-press/editorial-policies#Peer%20review>